# Sex Steroid Hormone Analysis in Human Tear Fluid Using a Liquid Chromatography—Mass Spectrometry Method

**DOI:** 10.3390/ijms232314864

**Published:** 2022-11-28

**Authors:** Alexandra Robciuc, Hanna Savolainen-Peltonen, Mikko Haanpää, Jukka A. O. Moilanen, Tomi S. Mikkola

**Affiliations:** 1Obstetrics and Gynecology, Helsinki University Hospital, University of Helsinki, FIN-00029 HUS Helsinki, Finland; 2Helsinki Eye Lab, Ophthalmology Department, Helsinki University Hospital, University of Helsinki, FIN-00029 HUS Helsinki, Finland; 3HUSLAB, Helsinki University Hospital, FIN-00029 HUS Helsinki, Finland

**Keywords:** tear fluid, steroid hormones, estrogen, testosterone, mass spectrometry, high performance liquid chromatography

## Abstract

The marked sexual dimorphism prevalent in inflammatory/autoimmune diseases is mostly due to sex hormone actions. One common eye disease that disproportionately affects women is dry eye. Thus, our aim was to optimise our highly sensitive liquid chromatography–tandem mass spectrometry method for steroid hormone quantification in tear fluid (TF). We used tears and matched serum samples from 10 heathy individuals. Estrone, estradiol testosterone, progesterone, androstenedione, and dehydroepiandrosterone, were quantified with an HPLC coupled with a Triple Quad 5500 MS. Estrone was measured in 80% of female and 20% of male TF samples (mean ± SD, 68.9 ± 62.2 pmol/L), whereas estradiol was undetectable in tears. Progesterone was identified in half of the female tear samples (2.91 ± 3.47 nmol/L) but in none of the male samples, whereas testosterone was quantifiable only in male tears (0.24 ± 0.1 nmol/L). TF hormone levels were, on average, from 1.4% to 55% of systemic values. Estrone, progesterone, and testosterone levels in tears correlated with the matching serum samples (*r* = 0.82, 0.79, and 0.85, respectively), but androstenedione and dehydroepiandrosterone showed no correlations. Our LC–MS/MS method could detect five out of the six steroid hormones studied in individual human TF samples and could therefore be used to analyse the role of sex steroids in eye diseases.

## 1. Introduction

Sex steroid hormones such as estrogen, progesterone, and testosterone account for differences in many physiological responses, such as the immune and inflammatory responses [1]. Moreover, changes in steroid hormone synthesis and secretion affect the manifestation of numerous chronic diseases. Steroid hormones in the eye impact tissue morphology, aqueous tear output, blink rate, and immune function, all fundamental in maintaining a healthy ocular surface [1,2,3,4]. One of the most common eye diseases, dry eye disease (DED), is highly sexually dimorphic, with female sex as a significant risk factor.

The current hypothesis is that steroid hormones act at the ocular surface primarily through intracrine synthesis [5,6,7,8]. This is plausible because the role of intracrinology has been highlighted during recent years in many tissues, such as adipose tissue, bone, and even eye [9,10,11,12]. Most of the previous studies, however, have measured circulating rather than local sex steroid concentrations [13,14] in connection with the signs and symptoms of eye diseases allowing for minimal mechanistical implications.

The development of mass spectrometric methods in tissues has opened new possibilities for the analysis of individual hormones or steroid hormone panels with high sensitivity and specificity in low volume/concentration samples [15,16,17,18,19]. Compared to blood, tears are a less complex body fluid, and the sampling is less invasive. An important limitation, however, is the reduced tear volume that can be safely sampled—10–15 µL from each eye at a time—and even less from an aged or dry eye. Therefore, sensitive and accurate MS techniques are the method of choice for this analysis.

Our aim was to optimise a highly sensitive LC–MS/MS method for the quantification of sex steroid hormones in tear fluid (TF). Furthermore, we aimed to compare local hormone concentrations to circulatory levels to determine if steroid hormones are synthesised locally in the eye or if they diffuse from the systemic hormone pool.

## 2. Results

We detected E_1_, P4, T4, A4, and DHEA in the TF samples. Only E_2_ and T4 were below the detection limit in the pooled female sample, which was used to optimise the analysis method (Table 1). There were no traces of hormones in either the water blank or the PBS control sample.

E_1_ could be assayed in 89% of the female (*n* = 9) and 20 % of the male (*n* = 5) samples in the individual sample analysis. TF-E_1_ levels were 60% lower than the corresponding serum concentrations: 48 (24–200) pmol/L in the TF vs. 160 (52–368) pmol/L in the serum (median (range), Table 2). P4 was quantifiable in 56% of the female TF samples (1.12 nmol/L (0.04–8.54 nmol/L)), but in no male samples. Female P4 TF concentrations were 70% lower than in serum. TF E_1_ and P4 in the samples obtained from females in two different menstrual cycle phases varied proportionally with the serum levels (*r* = 0.82 and 0.79, respectively; *p* < 0.01—Figure 1c,e).

T4 was quantifiable only in male tears (median (range); 0.2 nmol/L (0.1–0.4 nmol/L)), TF values being almost 100-fold lower than in serum. Serum and TF T4 levels likewise showed significant positive correlation (*r* = 0.85, *p* < 0.01, Figure 1d). A representative example of the T4 analysis in female TF was added as Appendix A.

The precursor hormone A4 was the only target molecule that could be quantified in all the TF samples (median (range); 1.2 (0.4–2.3) nmol/L), with more than three-fold lower concentrations than in serum (3.9 (1.33–11.5) nmol/L). A4 was significantly lower in female tears compared to male, whereas the finding was opposite in serum. However, no correlation existed between serum and TF values (Figure 1a,b). DHEA was measurable in 40% of the samples (*n* = 6), with values ranging from 0.8 to 2.8 nmol/L. It was below our detection limit in the one sample from a female using combined oral contraceptives. DHEA levels in tears were one order of magnitude lower than in serum (median 1.2 nmol/L vs. 19.1 nmol/L, respectively).

The chromatograms in Figure 2 show the LC–MS/MS separation and HPLC retention times (RT) of the steroid hormones in female and male samples. The chromatograms were generated by separate LC–MS/MS runs for increased sensitivity and specificity of the steroid hormone assay.

## 3. Discussion

We were able to quantify E_1_, P4, T4, A4, and DHEA from TF samples (20–50 µL) with minimal sample extraction using our highly sensitive LC–MS/MS method. We could observe a clear dichotomy between men and women for E_1_, P4, and T4; T4 was present only in males, while E_1_ estrone and P4 were almost exclusively found in female TF. DHEA levels did not differ between men and women. A4; however, was significantly lower in female tears compared to men, whereas the opposite occurred in the serum. Finally, tear fluid levels of E_1_, P4, and T4 correlated significantly with the corresponding circulating levels, but the precursor hormones A4 and DHEA showed no correlation.

The significant correlation between serum and TF E_1_, P4, and T4 could imply either diffusion or an active transport of the hormones from circulation. More than 90% of circulating sex hormones are bound to either serum albumin or sex hormone-binding globulin (SHBG) [20], while only a small fraction is free to enter cells and interact with their respective receptors. Thus, only non-bound steroid hormones can passively diffuse through plasma membranes, justifying the lower levels of these hormones in tear fluid or saliva [21]. TF steroid hormone levels were constantly lower than the corresponding serum levels in our study, but the ratio between tear and serum concentrations varied within a considerably wide range, from 1 to 50%. There are several explanations for the differences. An SHBG-receptor-mediated transmembrane transport emerged in recent years [22] in addition to the discovery of organic anion transport polypeptides, shown to bind steroid hormones [23]. SHBG has not been identified in tears to this date, and only about 0.1% of the serum albumin was shown to diffuse into tears [24]. One of the major tear proteins, lipocalin, however, has been shown to interact with numerous lipid species and could function as a serum-albumin replacement [25] and carry the neutral steroid hormones in the aqueous tears. A form of active transport or, conversely, local synthesis, is also supported by the significant difference between A4 in male and female tears, if for men the tear/serum ration is almost 50%, in women is less than 20%, on average. Additionally, the low levels of DHEA in tears may suggest that precursor hormones could be taken up by the nearby cells and locally metabolised into other sex steroids [8,26]. TF is the only supply of nutrients, minerals, and other active molecules for some ocular surface tissues, such as the cornea. Therefore, corneal tissue requires access to systemic DHEA or A4 for further local synthesis of active estrogens or T4. Cells at the ocular surface as well as adnexa contain all the steroidogenic enzyme mRNAs necessary for both androgen and estrogen intracrine metabolism [1]. Moreover, DHEA and A4 levels in TF did not correlate with serum levels, further supporting local uptake and possible metabolic use.

Quantification of sex steroids in tears has had very limited success to date with methodologies thwarted by the minute analyte concentrations and low sample volumes [27]. A previous study could quantify only three analytes from 14, and another was unable to detect five of seven analytes in 25% of their samples [24,28]. We could quantify five of the six hormones analysed. A4 was quantifiable in all samples, while DHEA was detected in all samples analysed except for one study subject that used oral contraceptives. Even with our low LOQ values, E_2_ could not be quantified in any of the samples, and T4 was quantifiable only in male samples, suggesting extremely low TF concentrations. T4 could not be detected even in the female pooled sample of tear fluid but could be easily detected in individual samples from males with volumes almost 10-fold lower (25–30 µL vs. 200 µL). This result is contrary to a previous study [27] showing T4 analysis in tears from women with/without dry eye, with levels comparable with our measurements in men. Our A4 and T4 tear fluid levels were similar to control values reported previously in female samples [27].

The clear advantage of our method is the optimised LC–MS/MS method and very low LOQs, while the minimal extraction needed diminishes sample loss. The smallest sample volume that provided results was 20 µL, which suggests that several hormones can be quantified from a single sample collection from healthy eyes. A limitation is that the requirement for separate runs for different hormones increases assay time and the sample volume used. It is also possible to measure sex hormones in tear extracts from Schirmer strips, a collection method suitable for both healthy controls and patients with ocular surface disease [27] or meibum lipid extracts. However, the present study used tear fluid collected with glass capillaries to avoid the extraction protocol, as well as skin or polymer contamination from the Schirmer’s strips.

This present study shows that our highly sensitive LC–MS/MS method can be successfully used for quantification of steroid hormones in individual samples of tear fluid. The correlations found between hormones in tears and circulation would suggest that TF hormones are mainly transported from serum. The marked difference between the tear/serum ratios for the hormones analysed as well as the lack of correlation with systemic levels for hormone precursors recommends a more in-depth analysis of sex hormone levels in local tissues and meibum before drawing such a conclusion. This information is essential in the process of finding novel therapeutic approaches for ocular diseases and dysfunctions.

## 4. Methods and Materials

### 4.1. Tear Fluid Sample Collection and Preparation

We first used a pooled sample of TF to optimise our LC–MS/MS assay for the steroid hormone analysis, after which we measured estrogen and androgen levels in TF from 10 healthy donors (five females of 31.0 ± 10 years and five males, 32.6 ± 12.5 years, mean ± SD). The study subjects had no ocular surface diseases or eye medication. One of the women used combined oral contraceptives and one had a levonorgestrel intrauterine system.

For the pooled sample we used 200 µL of tears from one female (39 years of age) collected with minimal conjunctival irritation in the span of one month, not more than 30 µL at a time from both eyes, using a 5 µL glass micro capillary tube (BLAUBRAND; intra MARK, Wertheim, Germany) positioned at the temporal corner of the lower eyelid. For the analysis of individual samples, we used 25 to 50 µL of TF from women (the 50 µL sample was collected in two consecutive days), while the sample size from the male volunteers varied between 20 to 30 µL. The collection procedure for four women was performed two times with a two-week interval. This was conducted with the intention of sampling during both the follicular and the luteal phases of the menstrual cycle. This protocol was not followed for the female volunteer with combined oral contraceptive use due to their suppressing effect on pituitary function and the resulting suppression of hormonal fluctuations. TF was centrifuged 5 min at 10,000 *g* to remove debris and stored at −80 °C until analysis. PBS, handled in parallel with the TF samples, acted as the control for the collection and analysis method. Blood samples were obtained for comparison on each day of TF collection. Blood samples were allowed to coagulate at room temperature for 30 min and then centrifuged to separate the serum, which was stored at −80 °C until analysis. All experiments were performed in accordance with the guidelines of the Declaration of Helsinki and the Ethical Committee of the Helsinki and Uusimaa Hospital District. Informed consent was obtained from all volunteers.

### 4.2. LC–MS/MS Analysis

TF and serum estrone (E_1_), estradiol (E_2_), progesterone (P4), testosterone (T4), androstendione (A4) and dehydroepiandrosterone (DHEA) concentrations were analysed by LC–MS/MS: Agilent 1200 HPLC (Agilent Technologies Inc., Santa Clara, CA, USA) coupled with AB Sciex Triple Quad 5500 mass spectrometer controlled by Analyst Software 1.6.2 (AB Sciex, Concord, ON, Canada). We adapted and optimised our previously described methods originally developed for serum or saliva to obtain adequate sensitivity and specificity for TF samples [15,16,17]. All the steps of the sample and calibrator preparation were performed using certified laboratory borosilicate glass vials (Agilent Technologies Inc.) and glassware to minimise the risk of the interfering contamination from plastics or from surroundings.

Assay calibrators of 0.0–1000 pmol/L E_1_ (Vetranal, Sigma–Aldrich, St. Louis, MO, USA), 0.0–1275 pmol/L E_2_ (Sigma–Aldrich), 0.0–50 nmol/L A4 (Sigma–Aldrich), 0.0–100 nmol/L T4 (Sigma–Aldrich), 0.0–100 nmol/L DHEA (Sigma–Aldrich) and 0.0–80 nmol/L P4 (IsoSciences) were prepared in water:methanol (1:1, *v*/*v*). To 200 µL of calibrators, 200 µL of serum and 20–50 µL of TF samples, 30 µL of internal standard (IS) containing 3 nmol/L 13C3-E_1_, 3 nmol/L 13C3-E_2_, 0.1 µmol/L 13C3-A4, 0.2 µmol/L 13C3-T4, 0.2 µmol/L 13C3-DHEA and 0.4 µmol/L 13C3-P4 (IsoSciences, Ambler, PA, USA) in water:methanol (1:1, *v*/*v*) were added. A serum sample, TF sample, or assay calibratorwith IS was extracted by 1 mL of diethyl ether (DEE). DEE phase was transferred into a vial and evaporated to dryness after centrifugation. 300 µL of 0.1% ammonia water (Sigma–Aldrich) and 1 mL of DEE were added to the residue of the serum sample extract, followed by a second extraction and evaporation of the DEE phase. The TF and serum sample or assay calibrators residues were dissolved in 160 µL of water:methanol (1:1, *v*/*v*).

Steroid hormones from calibrators and sample extracts were measured in four separate LC–MS/MS runs with the following selection of analytes and injection volumes: (1) E_1_ and E_2_, 100 µL; (2) A4 and T4, 5 µL; (3) DHEA, 40 µL; and (4) P4, 2 µL. Active hormone measurements were prioritised when volume was insufficient for all analyses. DHEA was not analysed in all TF samples for this reason. Chromatographic separation for all analytes was performed on a tandem column where a Discovery HS F5-3 column (2.1 × 100 mm, 3 µm; Supelco, Bellefonte, PA, USA) was coupled with a SunFire C18 column (2.1 × 50 mm, 3.5 µm; Waters, Milford, MA, USA). The mobile phase was a linear gradient consisting of ammonium fluoride in water (A) and methanol (B) at a flow rate of 300 µL/min. The gradient for E1 and E2 was 0 min 50% B, 4.5–10 min 100% B and 10.5–19 min 50% B; for A4 and T4: 0–3 min 50% B, 8–13 min 100% B and 13.5–22 min 50% B; for DHEA: 0 min 50% B, 3–9 min 100% B and 9.5–17 min 50% B; and for P4: 0–3 min 50% B, 8–15 min 100% B and 15.5–24 min 50% B where A was 40 µmol/L ammonium fluoride in water for E1 and E2, and 200 µmol/L ammonium fluoride in water for A4, T4, DHEA, P4 gradients.

E_1_, E_2_, and corresponding ^13^C3-labelled internal standards were detected for duplicate quantitation by the multiple reaction monitoring in the negative ion [M − H]^−^ mode, while P4, T4, A4, DHEA and respective internal standards were detected in the positive ion [M + H]^+^ modes using the following parent ions and selected transitions: E_1_ [M − H]^−^ m/z 269.1 to m/z 269.1 and m/z 145.0; E_2_ [M − H]^−^ m/z 271.2 to m/z 271.2 and m/z 183.1; A4 [M + H]^+^ m/z 287.1 to m/z 109.2 and m/z 97.2; T4 [M + H]^+^ m/z 289.1 to m/z 109.1 and m/z 97.0; DHEA as an ammonium adduct [M + NH4]^+^ m/z 306.1 to m/z 253.1 and m/z 213.3; P4 [M + H]^+^ m/z 315.1 to m/z 109.1 and m/z 96.9; 13C3-E_1_ [M − H]^−^ m/z 272.1 to m/z 272.1 and m/z 148.0; 13C3-E_2_ [M − H]^−^ m/z 274.2 to m/z 274.2 and m/z 186.1; 13C3-A4 [M + H]^+^ m/z 290.1 to m/z 112.2 and m/z 100.2; 13C3-T4 [M + H]^+^ m/z 292.1 to m/z 112.1 and m/z 100.0; 13C3-DHEA as an ammonium adduct [M + NH4]^+^ m/z 309.1 to m/z 256.1 and m/z 216.3; and with 13C3-P4 [M + H]^+^ m/z 318.1 to m/z 112.1.

Using Analyst Software 1.6.2 data processing tools for assay calibrators and analyte quantifications estimated limits of quantification (LOQ) were 5 pmol/L for E_1_ and E_2_, 10 pmol/L for A4, T4 and P4, 100 pmol/l for DHEA, where signal to noise ratios were S/N = 10 or higher. Specific blank sample chromatograms as well LOQ spectra with S/N values are presented in Appendix A, respectively.

### 4.3. Statistical Analysis

Statistical analysis was carried out using the IBM SPSS Statistics 25.0 software. We used a nonparametric test for inter-group comparisons and correlations: Mann–Whitney U for independent variables and Spearman’s correlation test. Data are presented (unless otherwise stated) as median (interquartile range) or median (range). The level of significance was *p* < 0.05.

## Figures and Tables

**Figure 1 ijms-23-14864-f001:**
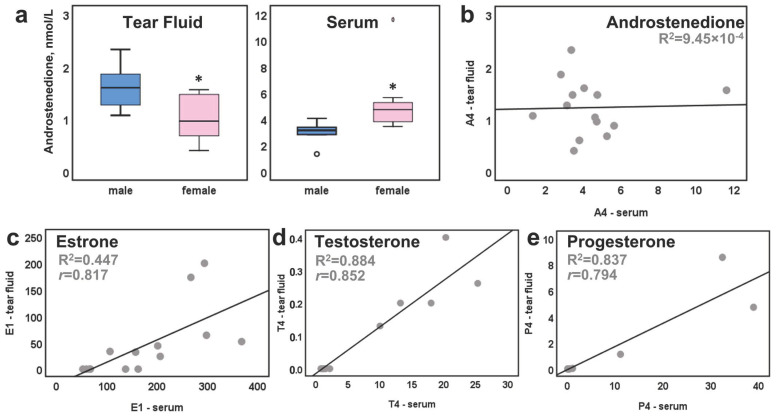
Graphical representation of TF and serum sex hormones. (**a**) A4 in female vs. male TF and serum, respectively; (**b**) correlation analysis between serum and TF for A4 levels, (**c**) E1, (**d**) T4, (**e**) P4 levels. A4, T4, and P4 are expressed in nmol/L, while E1 is pmol/L. *—statistical significance, *p* < 0.05.

**Figure 2 ijms-23-14864-f002:**
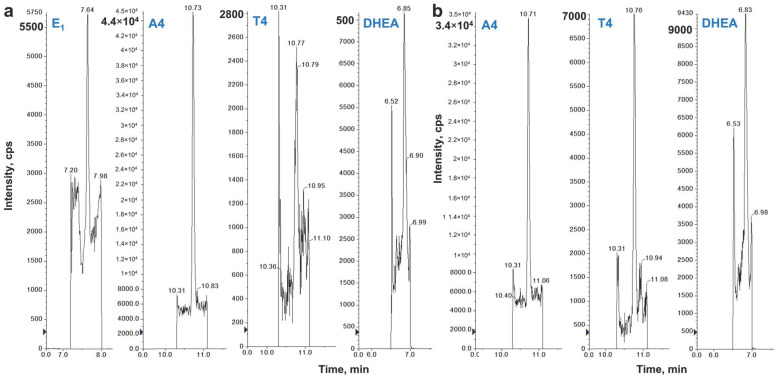
LC–MS/MS representative chromatograms for E1 (RT 7.6 min), E2 (RT 7.5 min), P4 (RT 11.6 min), DHEA (RT 6.8 min), A4 (RT 10.7 min), and T4 (RT 10.8 min) in TF from females (**a**) and males (**b**) E_1_—estrone; E_2_—estradiol; P4—progesterone; DHEA—dehydroepiandrosterone; T4—testosterone; A4—androstenedione. RT—HPLC retention time.

**Table 1 ijms-23-14864-t001:** Concentrations of sex hormones in a pooled sample of TF.

	E_2_ (pmol/L)	E_1_ (pmol/L)	P4 (nmol/L)	T4 (nmol/L)	A4 (nmol/L)	DHEA (nmol/L)
H_2_O blank	<5	<5	<0.05	<0.01	<0.01	<0.1
PBS blank	<5	<5	<0.05	<0.01	<0.01	<0.1
Tear fluid	<5	11	0.9	<0.01	0.5	0.6

E_1_—estrone; E_2_—estradiol; P4—progesterone; DHEA—dehydroepiandrosterone; T4—testosterone; A4—androstenedione; PBS—phosphate-buffered saline.

**Table 2 ijms-23-14864-t002:** Sex steroids in TF and serum.

		E_2_ (pmol/L)	E_1_ (pmol/L)	P4 (nmol/L)	T4 (nmol/L)	A4 (nmol/L)	DHEA (nmol/L)
	blank	<5	<5	<0.01	<0.01	<0.01	<0.1
female	TF 1 *n* = 5	<5	24 (0–173.4)	0.04 (0–8.5)	<0.01	0.68 (0.4–1.5)	0.8 (0–2.8)
	TF 2 *n* = 4	<5	48 (44–200.1)	1.12 (0–3.94)	<0.01	1.26 (0.8–2.0)	0.8 (0–1.2)
	Serum 1 *n* = 5	131 (15–698)	162 (52–294)	0.18 (0.1–1.1)	1.05 (0.7–1.4)	4.75 (3.5–5.6)	16.9 (14.8–26.4)
	Serum 2 *n* = 4	324 (248–434)	282 (201–368)	21.75 (0.4–38.9)	1.28 (1–2.1)	4.67 (3.4–11.6)	21.1 (14.6–30.1)
male	TF *n* = 5	<5	33.35	<0.01	0.2 (0.1–0.4)	1.6 (1.1–2.3)	2.01
	Serum *n* = 5	58 (21–80)	67 (59–137)	0.18 (0.1–0.4)	18 (10–25.3)	3.37 (1.3–4.1)	20 (6.1–27.4)

The data are expressed as median (interquartile range). E_1_—estrone; E_2_—estradiol; P4—progesterone; DHEA—dehydroepiandrosterone; T4—testosterone; A4—androstenedione; TF—tear fluid; TF 1 and serum 1—first sampling, while TF2 and serum 2 represent samples obtained after the two-week period.

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
