# Peer review of "Sex Steroid Hormone Analysis in Human Tear Fluid Using a Liquid Chromatography—Mass Spectrometry Method"

_ijms, 2022, doi:10.3390/ijms232314864_

Round 1
Reviewer 1 Report
The article is missing some essential information, such as structural information of the assay, mass spectra and chromatography of LOD, LOQ, and blank samples. Besides, isotope secondary mass spectra and chromatography should also be added.
Based on the figure 1, it is felt that the signal-to-noise ratio does not meet the requirements for quantitative analysis. Please provide the LOQ spectrum with the S/N ratio value. The quality and layout of the graph cannot meet the requirements of the journal, please revise.
It is difficult to obtain blood or tears without containing the endogenous components . In the article, the blank solution used for the standard curve are not blood or tear samples. Has the effect of matrix been examined? Have you considered the use of activated carbon adsorption to prepare a blank solution that is not good for the components to be tested?
Author Response
We greatly appreciate the reviewer’s valuable comments, and hope that we have answered all the concerns raised. We are constrained by the word limit (brief report, a total of 2500 words). Therefore, we suggest that the information we have now added in the methods section would be presented as supplementary data. We believe that the new data and revisions made in the text have improved our manuscript and hope that it is now acceptable for publication.
Our detailed reply is attached.

Reviewer 2 Report
The manuscript deals with sex steroid hormone analysis in human tear fluid using liquid chromatography-mass spectrometry.
Overall, the manuscript is interesting. However, some points are not clear.
Why the Authors used different gradient profiles for various compounds? Is it not possible to separate the mixture?
How do the Authors explain the strange peak profiles in Fig. 2? Peaks have few maxima, and they are really wide. Much broader than it should be.
Why retention time for E2 is listed as 7.5?
What was the MS mode for Fig. 2?
Why are there no units in Fig. 1?
Why the tandem column was used? Was it necessary?
Author Response
We thank the reviewer for the time conferred to our work and for the valuable comments. We hope that we have answered all concerns to the reviewer’s satisfaction and believe that the changes made improved the manuscript significantly. The word limit for brief report limits the extent of improvements. Therefore, we have included the required material as supplementary data and made the necessary changes to the methods and results chapters.
Our detailed reply is attached.

Round 2
Reviewer 1 Report
Accept in present form